# Evans’ Syndrome: From Diagnosis to Treatment

**DOI:** 10.3390/jcm9123851

**Published:** 2020-11-27

**Authors:** Sylvain Audia, Natacha Grienay, Morgane Mounier, Marc Michel, Bernard Bonnotte

**Affiliations:** 1Service de Médecine Interne et Immunologie Clinique, Centre de Référence Constitutif des Cytopénies Auto-Immunes de l’Adulte, Centre Hospitalo-Universitaire Dijon Bourgogne, Université de Bourgogne Franche Comté, 21000 Dijon, France; natacha.grienay@chu-dijon.fr (N.G.); bernard.bonnotte@u-bourgogne.fr (B.B.); 2Registre des Hémopathies Malignes de Côte d’Or, Centre Hospitalo-Universitaire Dijon Bourgogne, Université de Bourgogne Franche Comté, UMR 1231 Dijon, 21000 Dijon, France; morgane.mounier@u-bourgogne.fr; 3Service de Médecine Interne 1, Centre de Référence des Cytopénies Auto-Immunes de l’Adulte, Centre Hospitalo-Universitaire Henri Mondor, 94000 Créteil, France; marc.michel2@aphp.fr

**Keywords:** autoimmune haemolytic anaemia, immune thrombocytopenia, Evans’ syndrome

## Abstract

Evans’ syndrome (ES) is defined as the concomitant or sequential association of warm auto-immune haemolytic anaemia (AIHA) with immune thrombocytopenia (ITP), and less frequently autoimmune neutropenia. ES is a rare situation that represents up to 7% of AIHA and around 2% of ITP. When AIHA and ITP occurred concomitantly, the diagnosis procedure must rule out differential diagnoses such as thrombotic microangiopathies, anaemia due to bleedings complicating ITP, vitamin deficiencies, myelodysplastic syndromes, paroxysmal nocturnal haemoglobinuria, or specific conditions like HELLP when occurring during pregnancy. As for isolated auto-immune cytopenia (AIC), the determination of the primary or secondary nature of ES is important. Indeed, the association of ES with other diseases such as haematological malignancies, systemic lupus erythematosus, infections, or primary immune deficiencies can interfere with its management or alter its prognosis. Due to the rarity of the disease, the treatment of ES is mostly extrapolated from what is recommended for isolated AIC and mostly relies on corticosteroids, rituximab, splenectomy, and supportive therapies. The place for thrombopoietin receptor agonists, erythropoietin, immunosuppressants, haematopoietic cell transplantation, and thromboprophylaxis is also discussed in this review. Despite continuous progress in the management of AIC and a gradual increase in ES survival, the mortality due to ES remains higher than the ones of isolated AIC, supporting the need for an improvement in ES management.

## 1. Introduction

Evans’ syndrome (ES) was first described by Evans in 1951 [1] and is defined as the concomitant or sequential occurrence of immune thrombocytopenia (ITP) and autoimmune haemolytic anaemia (AIHA). ES-anaemia is an AIHA dues to warm antibodies that are usually of IgG isotype, exceptionally IgA, thus excluding cold agglutinins [2]. Autoimmune neutropenia (AIN) can also be part of ES, occurring in 15% cases in adults and 20% in children [3].

This review will focus on ES in adults. However, as clinicians managing adults will have to take care of patients diagnosed with ES during infancy, a specific paragraph is dedicated to ES in paediatrics. Moreover, ES in children can be part of a more complex clinical situation due to primary immunodeficiencies (PID) that will be discussed, as diagnosis of such syndromes might sometimes be suspected in adults.

Moreover, it must be specified that due to the rarity of the disease, there is almost no clinical trials comparing treatment modalities and that most of the recommendations that are given here are extrapolated from those of isolated ITP or isolated AIHA.

## 2. Evans’ Syndrome in Adults

### 2.1. Epidemiology

The knowledge about ES epidemiology in adults has been refined by a recent nationwide study in Denmark reporting on 242 patients managed for the last 40 years (1977–2017). The rarity of the disease was confirmed by an annual incidence of 1.8/million person-years, and an annual prevalence of 21.3/million persons [4]. When considering isolated AIC, ES represents 0.3–7% of AIHA and 2–2.7% of ITP [4,5,6].

In adults, autoimmune cytopenia (AIC) arise concomitantly in 30–57%, while ITP precedes AIHA in 27–44% [3,4]. The mean delay between the different AIC is of 3 years, but is highly variable [3,4].

ES is often diagnosed during the 5th–6th decades (median of 58.5 [55.9–61] in the Danish survey [4] and 55 +/− 33 in the French cohort [3]), with a slight female predominance (51–60%) [3,4], and runs a chronic course (>1 year) in more than 80%, with multiple relapses [3]. Importantly, ES is associated (secondary) to another disease in 27–50% cases, most particularly haematologic malignancies and systemic lupus erythematosus (SLE) in adults [3,4].

### 2.2. Diagnosis Procedure

#### 2.2.1. Diagnosis of ES

The diagnosis of ES relies on the concomitant or sequential diagnosis of AIC, but the delay between AIC occurrence is not a limiting factor.

AIHA is suspected in case of anaemia (haemoglobin <11 g/dL for female and <12 g/dL for male) associated with reticulocytosis and with markers of haemolysis, i.e., elevated lactate dehydrogenase, low haptoglobin and elevated indirect bilirubin, with a positive direct antiglobulin test (DAT) for IgG with or without complement (C3d) as cold agglutinins are excluded from ES [7].

ITP remains a diagnosis of exclusion suspected in case of rapid onset thrombocytopenia not related to liver diseases (cirrhosis and portal hypertension), splenomegaly (haematological malignancies, Gaucher disease,…), drug-related thrombocytopenia, bone marrow deficiency (myelodysplastic syndromes, haematological malignancies, metastatic cancer,…) or inherited thrombocytopenia [8]. Due to the lack of specificity or sensitivity of the different assays, the detection and identification of antiplatelet antibodies is still not recommended in routine practice and should be restricted to difficult cases [8]. However, when using direct Monoclonal Antibody Immobilization Platelet Assay (MAIPA), a sensitivity and a specificity of up to 81% and 98% have been reported, making this technique attractive for the diagnosis procedure [9].

AIN is suspected when facing a neutrophil count <1.5 G/L, after exclusion of other causes of neutropenia (drug-induced neutropenia; viral infections such as cytomegalovirus (CMV), Epstein Barr Virus (EBV), Human Immunodeficiency Virus (HIV), parvovirus B19, and influenza; myelodysplastic syndrome or leukaemia) as there is no specific test for its diagnosis [10]. Antineutrophil antibodies are quite difficult to determine in clinical practice as tests have not been standardized yet [11]. When antineutrophil antibodies are detected, they usually target Fc gamma receptor (FcγR), most particularly CD16 (FcγRIII) and more rarely CD32 (FcγRII), or the integrin CD11b or the complement receptor 1 (CR1/CD35) [12]. The diagnosis procedure of ES must exclude differential diagnoses and determine the primary or secondary nature of ES. Various explorations are recommended and reported in Table 1.

To avoid difficulties in the interpretation of biological tests, it must be kept in mind that some of these investigations must be performed before treating patients. Notably, intravenous immunoglobulins (IVIg) preclude the correct quantification of serum IgG, and immunosuppressants interfere with T and B cell phenotyping [13].

#### 2.2.2. Determining the Secondary Nature of ES

Due to the increased mortality in secondary ES compared to primary ES, diseases that are associated in up to 50% of ES in adults must be clearly identified at diagnosis [3].

##### Haematological Malignancies

ES has been reported to be associated with non-Hodgkin lymphoma (NHL) in 7% of a cohort of 68 adult ES, thus representing 15% of secondary ES [3], which is in line with the Danish nationwide cohort that showed that secondary ES due to haematological malignancies account for 21% of total ES [4]. Chronic Lymphocytic Leukaemia (CLL) is the most frequent lymphoproliferative malignancy associated with AIC, which occurred in up to 25% cases [14]. ES has been reported in up to 2.9% of a cohort of CLL patients, a frequency that is lower than isolated AIHA (7.3%) and isolated ITP (3.7%). In half of the cases, ES-AIC occurred simultaneously and were concomitantly diagnosed with CLL in 25% cases [15]. The median age at ES diagnosis was 66 years and 60% of patients were male. Interestingly, there was no difference regarding demographic and Binet stage between patients with or without ES. However, patients with ES had more frequently markers of poor prognosis, such as a higher expression of ZAP-70 (79% vs. 50%), an increase in unmutated IGHV (86% vs. 41%), more frequent del (17) (23% vs. 5%), and TP53 mutation (33% vs. 5%) which participate in the reduction of their overall survival [15].

Based on a prospective population-based registry of our area department, we confirmed that the frequency of ES associated with lymphoproliferative malignancies was low, representing 10 patients among a cohort of 3499 patients (0.29%) managed from 1995 to 2015. The frequency of ES depends on the underlying malignancies, occurring in 0.44% (4/911) of CLL, 0.43% (1/231) of T cell lymphoma, 0.24% (5/2078) of B cell lymphoma, while not observed among the 279 patients with Hodgkin lymphoma. Thus, the standardized incidence rate was 0.037/100,000 person-years for CLL, 0.041/100,000 person-years for B cell lymphoma and 0.007/100,000 person-years for T cell lymphoma. To characterize the clinical presentation of ES associated with lymphoproliferative malignancies, the 10 patients identified in the registry were gathered with 7 other patients managed in our institution during the same period, identified by the diagnosis-related group medical information system. The median age at diagnosis was 62 (interquartile range, IQR: 23–85) with 76% of male (13/17). The most frequent malignancies were CLL (41%, 7/17), marginal zone B cell lymphoma (17.5%, 3/17), or another low-grade B cell lymphoma (29.4%, 5/17). AIC occurred after the diagnosis of the lymphoproliferative malignancy by a median of 5 years (IQR:0.3–10 years) in 65% of the cases (11/17), occurred synchronously in 29% cases (5/17), while AIC preceded the diagnosis of the malignant hemopathy in only one case (6%). ES-AIC appeared simultaneously in most of the cases (59%, 10/17), while ES-thrombocytopenia preceded ES-anaemia in 29% of the cases (5/17), by a median of 0.5 year (IQR: 0.1–13.4). The treatment of the haematological malignancies was required in most of the cases (14/17, 82%), associated with a specific treatment of ES-thrombocytopenia in 88% (15/17) or ES-anaemia in 94% (16/17). A response was achieved in 70.5% for ES-thrombocytopenia (12/17) and 76.4% for ES-anaemia (13/17) after a median follow-up of 4.7 years (IQR: 0.4–27.9). After 5 years follow-up, 35% (6/17) of the patients had died. The death was related to the malignant hemopathy or its treatment in 4 cases (3 infections, 1 central nervous system localisation of lymphoma), while related to ES in 1 case (haemorrhagic stroke while in partial response of ES-thrombocytopenia) and of unknow origin for the remaining. At the time of death, 5 patients were under treatment for their malignant hemopathy and the remaining had relapsed. Concerning ES, 3 patients were responders for both anaemia and thrombocytopenia, 2 were responders for anaemia but not for thrombocytopenia and 1 was a non-responder for both.

In the diagnosis procedure of ES (Table 1), serum protein electrophoresis, protein immunofixation, and immunoglobulin concentrations are of interest to screen for abnormalities associated with lymphoproliferative malignancies such as monoclonal gammopathy, hypogammaglobulinemia, or polyclonal hypergammaglobulinemia. The phenotyping of circulating lymphocytes is useful to diagnose chronic lymphoid leukaemia or to identify blood circulating lymphoma cells. A bone marrow biopsy should be considered when a lymphoproliferative disorder is suspected, in case of adenomegaly, or splenomegaly disproportionated to haemolysis, hypogammaglobulinemia, or monoclonal gammopathy. In line with this, a CT scan of the chest, abdomen, and pelvis is required to detect lymph nodes and measure the spleen.

##### Autoimmune Disorders (AID)

AID have been reported in 18.2% of an adult ES cohort [4]. Systemic lupus erythematosus (SLE) is the most frequent AID associated with ES as reported in the largest cohort published to date including 68 ES: among the 34 (50%) secondary ES, 14 (20%) had an AID which was SLE in 10 (14%), the others suffering from Sjogren’s disease or antiphospholipid syndrome (2 for each) [3]. The association of ES and SLE was specifically studied in a Chinese cohort of more than 5,000 patients. ES was identified in only 27 patients, thus representing 0.47% of patients [16]. AIC occurred sequentially in more than half of the cases, with a diagnosis of SLE performed by a median of 3 years after AIC onset, ES and SLE being diagnosed concomitantly in only 30%. The phenotype of SLE patients with ES was different, with less lupus nephritis, more photosensitivity, and a more frequent polyclonal elevation of IgG when compared to SLE patients without ES [16]. Another cohort of 953 Brazilian SLE patients reported different results with 2.7% of patients with ES and a simultaneous diagnosis of ES and SLE in most of the cases (92%). As expected, SLE-associated ES occurred preferentially in women (90%) and was associated with another AID in 34% [17].

Thus, at the diagnosis of ES, it is recommended to measure antinuclear antibody to not undermine SLE (anti-dsDNA antibodies) or Sjögren disease (anti-SSA or SSB antibodies), while lupus anticoagulant assay and antiphospholipid antibodies should be considered in case of a previous history of thrombosis or obstetrical morbidity suggesting antiphospholipid syndrome (Table 1). In case of SLE-associated ES, the measurement of classical complement pathway activation (CH50, factors C3 and C4) is of interest as a marker of SLE activity.

##### Infections

Few papers reported on the association of ES with various viral infections (hepatitis C virus (HCV), Epstein Barr Virus (EBV), cytomegalovirus (CMV), and Varicella Zoster Virus (VZV)) that should be ruled out even though this situation seems rare. More recently SARS-CoV-2 has been reported as a potential cause of ES [18].

Due to their potential association with ES, specific viral serology testing and/or blood PCR for EBV and CMV should be considered when clinical exam is suggestive or in case of atypical lymphocytosis on the blood smear. Moreover, testing for Human Immunodeficiency Virus (HIV), HCV, and hepatitis B virus (HBV) must be performed to prevent reactivation or uncontrolled progression of infection upon immunosuppressive therapy (Table 1).

##### Primary Immunodeficiencies (PID)

Even though PID is most often diagnosed in children with ES, it should also be considered in adults when the clinical presentation is suggestive (consanguinity, family history of AIC or PID, recurrent infections, lymphoproliferation, lymphoma, hypogammaglobulinemia or polyclonal hypergammaglobulinemia (Table 1)). Due to the permanent improvement of the knowledge of genetic disorders, it is possible that unclassified ES will be associated with new genetic mutations in the future. Thus, clinicians managing patients who have developed ES during infancy should check for these PID if not previously done. Notably, ES could rarely reveals a PID in young adults [19], common immune variable deficiency (CVID) and, to a lesser extent, autoimmune lymphoproliferative syndrome (ALPS) being the most frequent PID associated with AIC [13], and are detailed in the section dedicated to paediatrics (Section 3.2. of the article).

##### ES and Pregnancy

One publication reported on the association of ES and pregnancy based on a systematic review of the literature, including 8 papers, consistent with 10 patients and 11 pregnancies [20]. ES is rarer than thrombotic microangiopathies (TMA) such as HELLP (Haemolysis with Elevated Liver enzymes and Low Platelet count) during pregnancy. The differential diagnosis will be supported by a positive DAT during ES, while negative during TMA with the presence of schistocytes on blood smear. From these 10 cases, ES was diagnosed for the first time during pregnancy and occurred sooner when due to a relapse of a previously known ES (8–14th weeks of gestation) than in newly diagnosed ES (14–38th weeks of gestation). Interestingly, none of the patients had neutropenia, which is usually observed in 15% of ES. Preeclampsia occurred during three pregnancies. The delivery was vaginal in most of the cases and as recommended during ITP, caesarean should only be considered for obstetrical reasons. Two stillbirths were observed, one probably due to thrombocytopenia leading to a cerebral haematoma, and the other due to severe haemolytic anaemia, which highlights the necessity to screen for congenital ITP or AIHA during the first days of life. Women were all treated with corticosteroids, at various dosages, as monotherapy for 4 patients. IVIg were required for 3 women. The delivery occurred between 32 and 40 weeks of gestation. Although the low number of patients and the various length of follow-up (2 months-8 years) preclude to draw firm conclusions, it seems that the course of the disease was favourable after the delivery, as most of the women did not require further treatments.

#### 2.2.3. Differential Diagnoses

##### Thrombotic Microangiopathies

Thrombotic microangiopathies (TMA) represent rare diseases with a devastating prognosis without treatment [21]. TMA are characterized by platelet aggregation in microcirculation, leading to thrombocytopenia, mechanical destruction of red blood cells (RBC), and various organ failures depending on the site of thrombosis. Thrombocytopenia associated with mechanical haemolysis and schistocytes on the blood smear are the hallmarks of the disease. However, in a cohort of 423 patients with thrombotic thrombocytopenic purpura (TTP), it has been shown that TTP could be misdiagnosed as an AIC in up to 20% [22]. The absence or a low level of schistocytes on blood smear at initial presentation and a weak positive DAT were responsible for such a pitfall. Despite a delay of four days in the diagnosis, the mortality did not increase. Patients were first treated with steroids and IVIg, which did not improve cytopenia. Thus, the assessment of schistocytes on blood smears must be repeated overtime and the diagnosis of ES should be promptly reconsidered when first line treatments are ineffective (Table 1). Of note, patients with SLE can develop ES or TTP during the course of the disease.

##### Anaemia Due to Bleeding Complicating ITP

Even though severe bleedings are rare during isolated ITP, they can lead to gastro-intestinal bleedings responsible for anaemia. In these cases, a careful examination of the full blood count will rule out ES as anaemia is normocytic and non-regenerative in case of acute bleeding or microcytic in case of chronic bleedings. Moreover, haemolytic parameters (haptoglobin, LDH, free bilirubin), if measured, will be in normal ranges, with a negative DAT.

##### Vitamin Deficiencies

Anaemia due to vitamin B12 deficiency is often associated with intramedullary haemolysis responsible for high level of LDH and low haptoglobin and sometimes the presence of numerous schistocytes on the blood smear, and can also be associated with thrombocytopenia [23]. The non-regenerative and megaloblastic (mean corpuscular volume >120 fL) characteristics of the anaemia, associated with low vitamin B12 blood levels, rapidly rule out the diagnosis of ES.

##### Myelodysplastic Syndromes

Myelodysplastic syndromes (MDS) are responsible for cytopenia that can affect multiple lineages, most particularly RBC and platelets [24]. ES is easily excluded as anaemia is non-regenerative and dysplastic features can be observed on blood smear. Bone marrow aspiration for cytologic examination and karyotyping will confirm the diagnosis of MDS (Table 1). In case of haemolysis associated with MDS, paroxysmal nocturnal haemoglobinuria (PNH) should be ruled out by flow cytometry as PNH clones have been observed in 1–2% of MDS (Table 1). ES has been unusually associated with MDS in case reports. In the largest cohort of 68 ES reported by Michel et al., only 2 (3%) patients subsequently developed MDS [3].

##### Paroxysmal Nocturnal Haemoglobinuria

Paroxysmal Nocturnal Haemoglobinuria (PNH) is a rare, acquired bone marrow failure due to somatic mutation in phosphatidylinositol glycan class A (PIGA), one of the genes involved in glycosylphosphatidylinositol (GPI) anchor biosynthesis. The complement inhibitors CD55 and CD59 are both GPI-anchored proteins that are deficient during PNH, thus leading to complement activation on the surface of RBC and to their haemolysis. Because PNH can be associated with aplastic anaemia, patients can also present with other cytopenia that can mislead to the diagnosis of ES. However, during PNH, TDA is negative and when aplastic anaemia is associated, the reticulocyte count is lower than what is observed during isolated AIHA. The diagnosis of PNH is confirmed by flow cytometry using a specific reagent (fluorescent aerolysin, FLAER) that binds to GPI and allows the determination of the deficit. Bone marrow aspiration and biopsy are of interest and show hypoplastic bone marrow in case of associated-aplastic anaemia (Table 1). Corticosteroids are poorly efficient during PNH, whose treatment relies on complement inhibitors such as eculizumab and bone marrow transplantation in case of associated-aplastic anaemia [25].

### 2.3. Clinical Management of Adulthood ES

Due to the rarity of ES, no clear therapeutic regimen has been established. However, treatments are mostly extrapolated from those commonly used for isolated ITP and isolated AIHA, and are summarized in Table 2.

#### 2.3.1. First Line Therapies

##### Corticosteroids

Corticosteroids represent the cornerstone therapy, used at a daily dose of 1 mg/kg of prednisone. The duration of treatment is determined by the AIC: 3–4 weeks with a brutal discontinuation or a rapid tapering over one week for ES-thrombocytopenia [26] and a slow tapering over six months for ES-anaemia [7]. Higher dosages of prednisone (up to 1.5 mg/kg) have been proposed to manage AIHA and by extension ES-anaemia; however, due to their side effects, corticosteroids should not be used for more than 3–4 weeks at this dosage and a second-line therapy should be rapidly considered in non-responder patients. In severe cases, notably life-threatening situations, pulses of methylprednisolone (up to 15 mg/kg/day) can be required.

Initial response rates are as high as 80% but the one-year-remission rate after corticosteroid as monotherapy is low for isolated ITP (20–30%) and for isolated AIHA (33%) [31,32]. Importantly, due to the autoimmune mechanism responsible for ES-neutropenia, corticosteroids and immunosuppressants should not be considered a contraindication in ES-neutropenia.

Dexamethasone (40 mg/day for 4 days) has been used for isolated ITP, leading to a faster response but a similar long-term response compared to prednisone [27,28]. No data are available for isolated AIHA, nor for ES-thrombocytopenia and ES-anaemia.

##### IVIg

Concerning ITP, IVIg should be restricted to patients with low platelet count (<30 G/L) associated with important bleeding symptoms, best assessed by using a bleeding score [63]. IVIg are usually used at 1 g/kg on day 1 and they could be repeated on day 3 if the platelet count remains below 30 G/L [26]. IVIg can be used solely as first line therapy when steroids are contraindicated or inefficient. In other cases, they are associated with corticosteroids allowing a quicker increase in platelet count [30]. Of note, IVIg represent only an emergency therapy that does not modify the natural history of the disease.

Only one study reported on IVIg during isolated AIHA and showed a low efficiency (12/37 patients (32%) increased their haemoglobin ≥2 g/dL, and only 15% achieved a partial response defined by haemoglobin ≥10 g/dL with an increase of haemoglobin ≥2 g/dL) [29], which precludes their routine use. Moreover, they might increase the risk of thrombosis that is already high during isolated AIHA, and might be similar in ES.

##### Transfusion Support

In case of symptomatic anaemia, RBC transfusions are required. The challenge during isolated AIHA and ES-anaemia is to not dismiss alloantibodies that can be masked by autoantibodies, notably in patients who have previously been transfused or women who have been pregnant. As autoantibodies usually lead to panagglutination of RBC by targeting antigens widely expressed on RBC such as glycophorin, protein band 3, and rhesus, specific techniques are needed to unmask potential alloantibodies [64]. These techniques mostly rely on autoadsorption as the previous technique using the dilution of serum allows the detection of alloantibodies in only 20% of the cases. Autoadsorption is based on the utilisation of RBC from the patient, previously eluted from bounded antibodies. The serum of the patient is then added to his RBC, leading to the fixation of autoantibodies. The adsorbed serum can then be tested for the presence of alloantibodies. The depth of the anaemia can limit this procedure by complicating the obtention of a sufficient amount of RBC. Of note, if the patient has been transfused in the past 3 months, the technique should not be performed as a low amount of transfused RBC can be sufficient to adsorb alloantibodies and leads to a false negative test. When autoadsorption is not doable, alloadsorption could be done, by using RBC obtained from various and specifically selected donors. However, this technique is time-consuming and requires a great expertise that limits its workability [64].

Platelet transfusion is not recommended during isolated ITP and by extension during ES-thrombocytopenia, due to the short half-life of platelets after transfusion and the fact that it does not improve outcome in most of the patients [52]. However, platelet transfusions are required in case of life-threatening bleedings, in association with immunomodulatory drugs, corticosteroids, and IVIg notably [8,53].

#### 2.3.2. Second Line Therapies

##### Rituximab

Rituximab is a drug of choice in ES as its efficacy has been clearly demonstrated in both isolated AIHA, with response rate of 75% at 1 year follow up [31,32] and in isolated ITP, with initial response rate in 60% and long-term remissions in 30% [33]. In ES, the initial response rate to rituximab was 82%, which dropped to 64% at one-year follow-up [3].

Data regarding the use of rituximab in secondary ES are scarce. One study specifically assessed rituximab in SLE-associated AIC in 71 patients, among which 11 had ES [65]. An overall response to rituximab was achieved in 60% cases, which is lower than the ones observed in cases of isolated AIHA or isolated ITP associated with SLE, respectively, of 87.5 and 91%. A complete response was achieved in 50% of ES as compared to 75 and 57%, respectively.

Concerning haematological malignancies-associated ES treated with rituximab, there is only one study that reported specifically on CLL [15]. Among the 25 patients, the response to treatment was available in only 72% cases. Half of the patients received only corticosteroids or IVIg, while the others were treated with chemotherapy including or not rituximab due to CLL progression. Response rate tends to be slightly higher when chemotherapy was used (ES-thrombocytopenia: 78% with 67% CR; ES-anaemia: 100% with 38% CR) compared to corticosteroids or IVIg alone (ES-thrombocytopenia: 71% with 42% CR; ES-anaemia: 87% with 25% CR). Unfortunately, rituximab was not used alone in this cohort. Thus, data are needed to determine whether it is an efficient treatment when ES is associated to haematological malignancies that do not specifically require treatments and whether its use as monotherapy modifies the long-term prognosis of the haematological malignancy.

##### Splenectomy

Splenectomy is an efficient treatment of both isolated ITP and isolated AIHA leading to response rates of 88% (66% complete response) [36] and 70% (40% complete response), respectively [7,37].

Data concerning splenectomy in ES are derived from small series showing response rates that are quite similar than those observed in isolated AIC with an initial response rate of 78–85%, with long-term response ranging between 42–62% [3,66]. Splenectomy should be avoided in case of ALPS and should be discouraged for patients with SLE, especially if they have positive antiphospholipid antibodies.

##### Immunosuppressants

As in isolated AIC, various immunosuppressants have been used in ES, mostly before the availability of rituximab. Cyclophosphamide, azathioprine, ciclosporin, or mycophenolate have been used, usually in association with corticosteroids, and allowed a response in 50–100% (Table 2). Nowadays, they should be restricted to patients who did not respond to corticosteroids, rituximab, and splenectomy, except for haematological-neoplasm-associated ES that requires chemotherapy. In case of ALPS, both splenectomy and rituximab should be avoided due to the increased risk of infectious complications. In a cohort of 30 children with refractory AIC, sirolimus appears to be of great interest as supported by a response rate of 100% (12/12) in ALPS patients [67]. Interestingly, sirolimus also triggered a quite good response (55.5%, 10/18) in the remaining patients with refractory AIC not due to ALPS, among which half of the eight patients with ES achieved a response [67]. To date, no data are available for ES in adults.

In the coming years, the use of immunosuppressants will probably decrease due to novel therapies that should be efficient in both isolated AIHA and isolated ITP, and by extension in ES, such as fostamatinib (a syk inhibitor that blocked phagocytosis), inhibitors of the neonatal Fc receptor, or inhibitors of the classical complement pathway [68,69].

##### Hematopoietic Stem Cell Transplantation

High dose chemotherapy followed by hematopoietic stem cell transplantation (HSCT) is rarely performed for AIC compared to other AID and is restricted to patients who are refractory to multiple lines of treatments. The results of a study assessing the tolerance and efficacy of autologous lymphocyte-depleted peripheral blood stem cell transplantation conducted by the National Institute of Health (NIH) including 14 patients (9 ITP and 5 ES) will not be detailed here as only 2 adult patients with ES were treated [70]. Briefly, the overall response rate was 57%, with a good tolerance of the procedure, although 2 patients died from causes unrelated to the procedure.

Data from the European Group of Blood and Marrow Transplantation (EBMT) that concerned 36 patients (among which 7 had ES, 12 isolated ITP, 7 isolated AIHA, 5 pure red cell aplasia, 2 pure white cell aplasia and 3 TTP) with 38 transplant procedures were first reported in 2004 [71]. Autologous transplant was performed in 27 patients, with 26 evaluable cases showing prolonged response in 9/26 (34.6%), transient response in 6/26 (23%), non-response in 7/26 (26.9%), death related to treatment in 3/26 (11.5%), or disease progression for 1 patient (3.8%). Among the nine patients who underwent allogeneic transplantation, 7 were evaluable, with a prolonged response in 5/7 (71.4%), the 2 others dying from complications related to the transplantation or to disease progression (one each). Overall, the progression free-survival was 45 +/− 21% for autologous transplantation and 78 +/- 28% for allogeneic transplantation.

In this cohort, 7 patients had ES: 2 received an autologous transplantation leading to prolonged response in one patient and transient response for the other, while 5 patients had allogeneic transplantation leading to prolonged remission in only one case, 2 being not evaluable and 2 dying from complications related to transplantation or disease progression.

Data from the EBMT were updated in 2008, with 65 transplantations (37 autologous and 28 allogeneics) performed in 59 patients. Considering all AIC, the 5-year survival was 79 +/− 14% for autologous and 58 +/− 25% for allogeneic transplantation.

Twelve patients with ES were treated, 4 with autologous and 8 with allogeneic transplant, but their specific evolution and follow-up are not provided in the article [71].

Thus, considering that a sustained response could be achieved in only up to 25% of the patients, in our opinion, HSCT should be restricted to patients who are refractory to multiple line therapies and those that could not participate in clinical trials assessing new innovating drugs.

##### Bone Marrow Stimulating Agents: Thrombopoietin Receptor Agonists (TPO-RA) and Erythropoietin

The efficiency of TPO-RA in ES is only supported by case reports [61,62]. TPO-RA have clearly demonstrated their efficiency in isolated ITP, with response rate of 70–80% [59,60]. However, they are associated with an increased risk of thrombosis, notably in patients with cardiovascular risk factors, antiphospholipid syndrome, who underwent splenectomy or upon corticosteroids or IVIg therapies [72]. Considering the inherent risk of thrombosis during isolated AIHA [54,57,73], that can be extrapolated to ES-anaemia, TPO-RA should therefore be considered with caution in patients with ES and active haemolysis but they can be helpful for managing severe active ES thrombocytopenia without simultaneous ES-anaemia.

It has recently been shown in a multicentric retrospective international study including 51 patients with isolated AIHA that erythropoietin (EPO) could be of interest [58]. Most of the patients had received at least one therapeutic line and EPO was started after a mean course of 2 years because of a non-response to treatment in two-thirds of the cases. An increase in haemoglobin level of at least 2 g/dL was achieved in 70% cases, after 2 weeks in half of the patients. Interestingly, EPO could be discontinued in one-third of the responders. Of note, thrombotic events occurred in only two patients. Thus, considering the results observed in isolated AIHA, EPO seems to be efficient and well-tolerated and could be transiently considered for ES-anaemia in patients who do not achieve a correct response upon immunomodulatory medications.

##### Anticoagulation

It is now clearly established that isolated AIHA enhances the risk of thrombosis [54,57,73], most particularly when the disease is active, with a 7.5-fold increase during the three months following diagnosis. Even though, there are no clear guidelines regarding anticoagulation prophylaxis during isolated AIHA, experts recommend to consider thromboprophylaxis for inpatients in the active stage of the disease, taking into account their general risk factors for venous thromboembolic events (VTE) [7]. Indeed, thromboprophylaxis seems to decrease the occurrence of VTE, as reported in a monocentric study showing 5 VTE during 15 isolated AIHA exacerbations when prophylactic anticoagulation was not used, as compared to 1 VTE among 21 exacerbations when prophylaxis was given [55].

In our opinion, this approach could be extrapolated to ES, except in case of profound thrombocytopenia. Thus, we recommend prophylactic anticoagulation by low molecular weight heparin for ES patients that are hospitalized or with one general VTE risk factors such as age above 70, past history of VTE, reduced mobility, active cancer, known thrombophilia, recent surgery or trauma, acute infection, heart or respiratory failure, and who have an active ES-anaemia with a platelet count above 50 G/L.

### 2.4. Management of ES during Pregnancy

The management of ES during pregnancy is challenging as most of the drugs that are usually used in ES are not recommended, notably rituximab and TPO-RA, even though few reports showed favourable outcomes [74,75].

During pregnancy, corticosteroids remain the cornerstone therapy due to their high efficiency and short delay of action.

IVIg are also useful to treat ES-thrombocytopenia but are not recommended for ES-anaemia.

Azathioprine can be efficient on both ES-thrombocytopenia and ES-anaemia and could be maintained in case of ES prior to pregnancy. However, due to its long delay of action, azathioprine is of poor interest in case of ES emerging during pregnancy.

Splenectomy can be efficient on both ES-anaemia and ES-thrombocytopenia but is challenging during pregnancy and should be performed during the second trimester when needed.

### 2.5. Complications

#### 2.5.1. General Considerations

The median survival of patients with ES has increased over time and is around 7 years. Importantly, the survival is poorer in secondary ES compared to primary ES (1.7 vs. 10.9 years) [4], with a 5-year survival around 75% [3], which drops to 38% in secondary ES [4]. Overall, the survival is lower than in isolated AIC (8.7 years for isolated AIHA and 12.7 for isolated ITP), and far lower than the general population (21.1 years) [4]. Importantly, 30% of deaths occur within the first year of diagnosis, with an adjusted hazard ratio of death of 12.7 at 1 year, 2.3 between 1–5 years, and 1.5 between 5–10 years [4].

The causes of death are bleedings, with a similar frequency than observed in isolated ITP on the same period, and haematological neoplasms, notably for secondary ES [4].

An increase in mortality was also observed in a large Italian cohort of AIHA with an adjusted hazard ratio of death of 6.8 (95%CI: 1.99–23.63) for ES compared to isolated AIHA [6]. In this study, ES was shown to be a risk factor of death during AIHA, other risk factors being infections, acute renal insufficiency, previous splenectomy, and the need for more than three lines of therapy [6].

#### 2.5.2. Management of Life-Threatening Complications

Although life-threatening complications are rare during ES, they need prompt recognition and management. ES-thrombocytopenia responsible for life-threatening haemorrhage, i.e., intracranial, visceral haemorrhage, or gastrointestinal bleedings responsible for profound anaemia should be managed as recommended in isolated ITP, although these recommendations are based on expert committee reports, opinions, or clinical experiences (grade C) [8].

High dose of corticosteroids (daily pulses of methylprednisolone at a maximum dosage of 15 mg/kg per day for 3 days and no more than 1g/day) are used in association with IVIg (1 g/kg/day during two consecutive days) [8].

In order to obtain a rapid haemostasis, platelet transfusions are recommended in situation of severe uncontrolled bleeding and can be repeated every 8 h until bleeding resolved, and immunomodulatory drugs are efficient. In a retrospective review of 40 patients treated with IVIg and platelet transfusions, with a mean pre-treatment platelet count of 10 G/L, an increase to 55 G/L and 69 G/L was observed at day 1 and 2, respectively [53]. After 1 day, the platelet count was >50 G/L in 62.7% of the patients with a control on bleeding in all patients.

Weekly infusions of vinca alkaloids (10 mg vinblastine or 1 mg/m^2^ vincristine) are also recommended. In a prospective study enrolling 35 patients with isolated ITP who were not responders to corticosteroids and IVIg, the combination treatment of pulses of high dose methylprednisolone with IVIg and vinca alkaloids showed a response in 71% [76].

Recently, the French referral centre for AIC showed in a retrospective study that weekly high doses of romiplostim (10 µg/kg) added to corticosteroids and IVIg was of interest to improve the response rate [77]. Among the 30 patients included, 20 patients who received vinca alkaloids associated with romiplostim were compared to an historical cohort of 22 patients receiving vinca alkaloids without TPO-RA. All patients had severe bleeding with an absence of response to steroids and IVIg. Although the response rates were not significantly different at day 7 (70% vs. 48%), a complete response was achieved more frequently when romiplostim was associated with vinca alkaloid (60% vs. 29%), and both response and complete response rates were higher at day 14 (80% vs. 43% and 70% vs. 17% respectively). However, this strategy was associated with an increased risk of thrombosis that occurred in two patients receiving high dose of romiplostim (1 deep vein thrombosis with pulmonary embolism with a platelet count of 629 G/L and 1 ischaemic stroke with a platelet count of 239 G/L).

Despite its long delay of action (3 to 4 weeks), early administration of rituximab must also be considered in life-threatening situations in order to shorten the critical condition [8].

Rarely, emergency splenectomy should be considered with a response usually rapidly observed during the days following surgery [8].

#### 2.5.3. Life-Threatening Haemolysis

Data concerning the management of life-threatening haemolysis are scarce, particularly in ES, and the recommendations are based on isolated AIHA.

Due to the delay of effectiveness of immunomodulatory drugs (steroids, rituximab, immunosuppressants) used to manage isolated AIHA and by extension ES anaemia, RBC transfusions remain essential to avoid severe hypoxemia. As alloantibodies can account for up to 30% of patients with AIHA previously transfused or who had a pregnancy [78], in emergency situations, it is recommended to transfuse ABO-, Rhesus-, and K-matched blood [51,79].

High doses of methylprednisolone are recommended as previously mentioned for ITP, based on expert recommendations [51].

Rituximab should also be performed as early as possible to shorten the life-threatening period.

Despite few data concerning the efficiency of IVIg in isolated AIHA, with a response observed in one-third of patients [29], it can be discussed in this specific emergency situation.

To remove pathogenic autoantibodies, plasmapheresis has been used but with contradictory results in severe isolated AIHA. A review reported on plasmapheresis as adjunctive therapy with a response observed in 4/6 of isolated AIHA and 4/4 of ES [49]. A monocentric case-control study assessed the gain in haemoglobin in patients with severe AIHA, including eight patients with warm AIHA and two with cold agglutinins [48]. All patients required RBC transfusions, and five patients were treated with plasma exchanges. Five days after transfusion, neither the haemoglobin levels nor the gain in haemoglobin were different in the two groups. Thus, the American society for apheresis states that there is an unestablished role for apheresis in severe warm AIHA with a necessity of individualized decision due to the low-quality evidence available [50].

Inhibition of the classical complement pathway represents a new therapeutic approach in AIC. The efficacy of eculizumab, a C5b inhibitor, has been reported in few case reports of severe isolated AIHA [80,81] and could be considered in case of life-threatening ES-anaemia.

## 3. Evans’ Syndrome in Paediatrics

### 3.1. Epidemiology

In children, ES is also a rare situation, as reported in a recent Danish nationwide population-based study that identified 159 AIC among which 21 were ES, between 1981 and 2015 in children less than 13 years of age [82]. The incidence had risen from 0.5/million person-years between 1981 and 1990, to 1.2/million person-years between 2006 and 2015. The prevalence has also shown a marked increase, from 6.7/million persons in 1990 to 19.3/million in 2015. Thus, the incidence and prevalence of ES in children are quite similar to the ones in adults. In children, ES represents 11.7% of isolated AIHA and 0.7% of isolated ITP [82].

In this study, the mean age at diagnosis was 4.7 years (3.3–6), which is close to the one reported in the largest cohort of 156 paediatrics ES (<18 year-old) prospectively included since 2004 in 26 French centres with a mean age at diagnosis of 5.4 years (0.2–17.2) [34]. Contrary to other AID and for unexplained reasons, ES more frequently affects boys with a sex ratio of 1.5–2 [34,82], which differs from adults (sex ratio of 0.7–1) [3,4]. Similarly to what is observed in adults, AIC occurred concomitantly in 46%, while thrombocytopenia was the first manifestation in 29% and anaemia in 25%, with a mean delay of 2.4 years (0.1–16.3) between AIC [34].

The frequency of secondary ES is difficult to determine in children. In the French cohort, only 30% of ES were considered as primary and only 8% were associated with SLE [34]. Of note, no haematological malignancies were observed, which is in contrast to the Danish nationwide study that reported haematological malignancy-related ES in 19% (4/21 cases) [82]. As a comparison, ES is associated with haematological malignancies in 15–21% of cases in adults [3,4].

Thus, contrary to what is observed in adults, a specific genetic background of PID should be considered in the majority of ES in children (detailed in paragraph 3.2.2.). Moreover, the early-onset of SLE also raises the question of interferonopathies, diseases that are due to type-I interferon dysregulation and that mimic SLE [83]. To date, there is no clear association between interferonopathies and ES, but this will need further investigations.

### 3.2. Primary Immunodeficiencies Associated with Evans’ Syndrome.

#### 3.2.1. General Considerations

Based on the data of the French cohort including 156 children with ES, after a mean follow-up of 6.5 years (0.1–28.8), 30% were considered as primary and 10% were proven to be secondary, i.e., due to SLE (*n* = 13, 8.3%) or to ALPS (*n* = 3, 2%).

Thus, 60% of ES were considered to be associated to a still undefined-genetic background due to their association with other immune abnormalities such as polyclonal lymphoproliferation, hypogammaglobulinemia, or other AID [34]. This was further confirmed by another study that investigated genetic disorders during ES: 80 patients of the cohort underwent genetic analyses leading to a genetic diagnosis in 65% (*n* = 52), most of them having a pathogenic mutation in genes known to be involved in PID (*n* = 32) [84]. The PID associated with the most frequent pathogenic mutations (*FAS* mutation in ALPS, CTLA4 and LRBA deficiency) will be detailed below. Other mutations were less frequent and involved *STAT3* (its gain of function is responsible for AIC, lymphoproliferation, enteropathy, interstitial lung disease, thyroiditis, diabetes, and postnatal growth failure) [85,86], *PIK3CD* (to date, the gain of function of phosphoinositide 3-kinase δ has been associated to recurrent ears and respiratory infections leading to bronchiectasis, susceptibility to herpes group virus infections, lymphopenia due to increased activation-induced cell death, low serum IgG_2_ and high serum IgM levels) [87], *CBL* and *KRAS* (that are implicated in RAS-associated autoimmune leukoproliferative disorder (RALD) responsible for splenomegaly, AIC, monocytosis, pericarditis and skin manifestations) [88,89], *RAG1* (its loss of function being associated with refractory AIC, granulomatous disease and inflammatory skin disorders) [90,91], and *ADAR1* (to date, its mutation is responsible for Aicardi-Goutières syndrome that is associated with increased production of interferon-α and is responsible for clinical manifestations affecting the brain and the skin that mimic congenital viral infection) [92]. For 20 patients, a probable pathogenic variant was identified, affecting various genes coding for cytokine receptors (*TNFR2*, *IFNAR1*, *TGFBR2)*, for molecules involved in intracellular signalling (*JAK1, JAK2, PLCG2, CARD11, PTPN11*), for proteins involved in apoptosis pathways (*RIPK2, APAF1*), or for transcriptional factors (*IKZF1, IKZF2, NFATC1*) [84]. Importantly, children displaying PID-related mutations display a more severe disease, with an increased risk of death, while children without these mutations have more frequently other autoimmune or systemic inflammatory manifestations, and require more lines of treatment [84].

#### 3.2.2. Description of the Most Frequent Genetic Disorders Associated with ES

##### Common Variable Immunodeficiency (CVID)

CVID is the most frequent PID characterized by a deficit in the humoral immune response responsible for a low level of immunoglobulins, most particularly IgG and IgA. CVID patients have recurrent bacterial infections, mostly of the ears, nose, throat, and respiratory systems. They can also display lymphoproliferation (either benign or lymphoma) and are more prone to AIC. In an American registry involving 990 CVID patients, AIC accounted for 10.2%, with ES being diagnosed in 1.6%. Importantly, AIC were associated with non-infectious CVID complications such as interstitial lung disease, enteropathy, hepatic disease, lymphoproliferation, and granulomatous disease that are known to worsen the prognosis [93]. Infectious complications are usually easily managed by immunoglobulin substitution that is not efficient to prevent AIC that required immunosuppressive therapies such as corticosteroids, rituximab and rather rarely splenectomy [94]. Thrombopoietin receptor agonists (TPO-RA) are of particular interest to manage CVID-associated ITP as they act by stimulating the production of platelets without immunosuppression.

##### Autoimmune Lymphoproliferative Syndrome (ALPS)

ALPS is a rare disease generally revealed during the first decade, associating lymphoproliferation (adenomegaly, hepatosplenomegaly), AID, mostly AIC, and is associated with an increased risk of malignancies [43]. Biological tests show a polyclonal increase of gammaglobulins and high level of vitamin B12. Suspicion of ALPS should lead to the quantification of circulating double negative T cells (CD3^+^TCRαβ^+^CD4^−^CD8^−^) that are increased, and to the measurement of soluble FasL, IL-10 and IL-18 that can be increased. ALPS is due to various mutations of the FAS/FAS ligand pathway (*FAS* (also known as *TNFRSF6*), *FASLG,* and *CASP10* (caspase 10)) leading to inefficient FAS-mediated apoptosis of T cells responsible for the survival of activated T cells and the emergence of autoreactive T cells. When performed, lymph node biopsy shows reactive follicles. The diagnosis criteria were refined in 2010 and are based on the associations of the two required criteria (1. Chronic, i.e., more than six months, non-malignant, non-infectious lymphadenopathy or splenomegaly or both and 2. elevated CD3^+^TCRαβ^+^CD4^−^CD8^−^ double negative T cells, i.e., ≥1.5% of total circulating lymphocytes or ≥2.5% of T cells, in the setting of normal or elevated lymphocyte count) with at least one accessory criterion, among primary criteria (1. defective lymphocyte apoptosis observed in 2 separate assays, 2. somatic or germline pathogenic mutation in *FAS*, *FASLG* or *CASP10*) and secondary criteria (1. Elevated plasma sFASL levels (>200 pg/mL) or elevated plasma interleukin-10 (>20 pg/mL) or elevated serum or plasma vitamin B12 levels (>1500 ng/L) or elevated plasma interleukin-18 levels (>500 pg/mL), 2. Typical immunohistological findings as reviewed by an experienced haematopathologist, 3. AIC (AIHA, ITP or AIN) and elevated IgG levels (polyclonal hypergammaglobulinemia, 4. Family history of a non-malignant, non-infectious lymphoproliferation with or without autoimmunity) [95]. Similar phenotypes can also be observed due to mutation of *CASP8* (Caspase 8 deficiency state, CEDS) or *NRAS* (RAS-associated autoimmune leukoproliferative disease, RALD) [95]. ALPS patients have a 50-fold increased risk of Hodgkin lymphoma and a 14-fold increased risk for Non-Hodgkin lymphoma [96].

Treatment of ALPS is not always required, and a wait-and-watch strategy can be applied. When needed, corticosteroids could be used. AIC during ALPS are often refractory and require second line therapies [96]. In a retrospective cohort of 16 children with ES, either primary (*n* = 5) or associated with ALPS (*n* = 11), mycophenolate mofetil led to a high complete response rate that was similar in ALPS-associated ES (82%) and primary ES (80%) [42]. Similarly, sirolimus has shown its high efficiency by inducing a clinical and biological response in 12 children with ALPS-associated refractory AIC [67]. Importantly, splenectomy should be avoided in ALPS due to the high risk of infections. Similarly, rituximab, which is often used to treat AIC, should be avoided in ALPS-associated AIC as it has been associated with profound and persistent hypogammaglobulinemia [19].

##### CTLA-4 and LRBA Deficiency

CTLA-4 is expressed on activated T cells and competes with the activating receptor CD28 for binding to the costimulatory molecules CD80 and CD86 expressed on antigen-presenting cells, thus leading to the inhibition of T cells. LRBA is a molecule involved in the intracellular recycling of CTLA-4, its deficiency thus decreasing the expression of CTLA-4. Deficiencies of CTLA-4 or LRBA lead to a quite similar phenotype associating lymphoproliferation, lymphoid infiltration of various organs (enteropathy, infiltrative lung diseases, encephalitis), AID, notably AIC, predisposition to infections due to hypogammaglobulinemia and to lymphoma [97,98]. Despite a median age at onset of 11, the first symptoms related to CTLA-4 deficiency occurred after 18 in a quarter of a cohort composed of 133 subjects of 54 unrelated families [99]. Abatacept, a fusion protein of the extracellular domain of CTLA-4 and Fc domain of immunoglobulins is of particular interest in these patients [98].

### 3.3. Treatment and Prognosis of ES in Paediatrics

Of the 156 children of the French cohort, 69% required at least one second line treatment, and almost half needed more than one second line therapy. First line therapies relied on steroids and IVIg for ES thrombocytopenia, and steroids for ES anaemia. The main second line therapies were rituximab (31%), azathioprine (15%), splenectomy (12%), cyclosporin (9%), and mycophenolate mofetil (2.5%). At the last follow-up, 74% of the children had a complete response, while 10% died from infections (70%), that were favoured by ES treatments, while death was related to ES, most particularly bleeding due to uncontrolled thrombocytopenia for the others (30%) [34]. This underlines the high mortality rate in children with ES, which was confirmed in the Danish study showing hazard ratios of 22.3 [4.3–115], 11.8 [3.2–44] and 2.5 [0.3–21] as compared to the general population, isolated AIHA and isolated ITP, respectively [82].

## 4. Conclusions

ES is a rare combination of AIHA and ITP that is associated in 50% of adult cases with various diseases such as SLE, haematological malignancies, or PID, the latter being the most frequent in children. Its prognosis is poorer than the one of isolated AIC and is particularly worse when associated with haematological malignancies. Its management is mostly empirical and extrapolated from guidelines for both isolated AIHA and isolated ITP. Corticosteroids remain the first line therapy with a short course duration for ES-thrombocytopenia and of six months for ES-anaemia. Second line treatments are usually required and the ones that are efficient in both isolated AIHA and isolated ITP such as rituximab, immunosuppressants, and splenectomy are recommended. In specific situations such as ALPS, mycophenolate mofetil or sirolimus should be preferred. Treatments that could be required for managing isolated ITP and that are associated with an increased risk of thrombosis such as IVIg and TPO-RA should be used with caution in ES as ES-anaemia probably increases the risk of thrombosis, as observed in isolated AIHA.

## Figures and Tables

**Table 1 jcm-09-03851-t001:** Recommended investigations during Evans’ syndrome diagnosis procedure.

Diagnosis of Evans’ Syndrome
- Complete blood count - Reticulocyte count - Haptoglobin, LDH, indirect/free bilirubin - Direct Antiglobulin Test - Monoclonal Antibody Immobilization Platelet Assay (MAIPA) (not systematic, of potential utility if antiplatelet antibody determination is required) - Antineutrophil antibodies against CD16/FcγRIII, CD11b, CD35/CR1, CD32/FcγRII (not systematic, of potential utility)
**To exclude differential diagnosis and determine the secondary nature of ES**
- Blood smear * - Viral tests (HIV, HCV, HBV, EBV, CMV, parvovirus B19) - Serum protein electrophoresis, protein immunofixation and immunoglobulin concentrations - Circulating lymphocyte phenotyping - Flow cytometry for paroxysmal nocturnal haemoglobinuria clone detection * - Antinuclear antibodies and anti-dsDNA antibodies - Lupus anticoagulant assay and antiphospholipid antibodies - Bone marrow aspiration and karyotyping * - Bone marrow biopsy - CT scan of the chest, abdomen and pelvis - Genetic explorations

*: exams useful to exclude differential diagnosis.

**Table 2 jcm-09-03851-t002:** Treatment approaches of Evans’ syndrome in adults.

Treatment	AIHA/ES-Anaemia		ITP/ES-Thrombocytopenia		References
	Dosage/Recommendations	Response	Dosage/Recommendations	Response	
Corticosteroids	Prednisone 1 mg/kg/day (up to 1.5 mg/day) for 3–4 weeks, progressive tapering over 6 months	Initial: 80% Prolonged: 33%	Prednisone, 1 mg/kg/day for 3–4 weeks	Initial: 60–80% Prolonged: 20–30%	[3,8,26]
			Dexamethasone, 40 mg/day, 4 days	Initial: 80% Prolonged: 20–30%	[27,28]
	Methylprednisolone 15 mg/kg/day for 3 days (no more than 1 g/day) Recommended for life-threatening situation	Unknown	Methylprednisolone 15 mg/kg/day for 3 days (no more than 1 g/day) Recommended for life-threatening situation	Unknown	[7,8]
IVIg	0.4 g/kg/day, 5 days	Initial: 32%	1 g/kg/day, 2 days	Initial: 90%	[8,26,29,30]
Rituximab	375 mg/m^2^/week for 4 weeks or 1000 mg Day1 & 15	60–75%	375 mg/m^2^/week for 4 weeks or 1000 mg Day1 & 15	40–60%	[3,31,32,33,34,35]
Splenectomy	To be avoided in ALPS	70%	To be avoided in ALPS	88%	[7,36,37]
Azathioprine	2–2.5 mg/kg/day (of interest for pregnancy)	56–71%	2–2.5 mg/kg/day (of interest for pregnancy)	45%	[3,6,34,37,38]
Cyclophosphamide	1–2 mg/kg/day (50–200 mg/day)	70%	1–2 mg/kg/day (50–200 mg/day)	60%	[3,6,26,39]
Cyclosporin	2.5 mg/kg twice per day (of interest for pregnancy)	58%	1.5–2.5 mg/kg twice per day (of interest for pregnancy)	44–55%	[3,6,26,34,40,41]
Mycophenolate	500–1000 mg twice per day	25–100%	500–1000 mg twice per day	45–60%	[3,6,26,34,37,42,43,44,45]
Vinka-alkaloid	ND	ND	Vinblastine: 10 mg/week Vincristine: 1–2 mg/week	Initial: 41–86%	[8,26,46,47]
Plasma exchange	To be considered in life-threatening haemolysis as adjunctive therapy	Not known	Not recommended		[48,49,50]
Transfusion	ABO-, Rh-, K- matched RBC		Platelets are not recommended except in life-threatening haemorrhage combined with immunomodulatory drugs		[7,8,51,52,53]
Anticoagulation	Thromboprophylaxis with low molecular weight heparin recommended for in-patients with acute exacerbation		Stop if platelet count <50 G/L		[54,55,56,57]
Bone marrow stimulating agents	Erythropoietin: to be considered in patients with unappropriated reticulocyte count or insufficient response upon immunomodulatory drugs Increased risk of thrombosis: to avoid in patient with risk factors	70–80%	Thrombopoietin receptor agonists: to be considered if ES-thrombocytopenia is the main problem Increased risk of thrombosis: to avoid if active haemolysis or thrombosis	70–80%	[58,59,60,61,62]

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
