# Peer review of "Evans’ Syndrome: From Diagnosis to Treatment"

_jcm, 2020, doi:10.3390/jcm9123851_

Round 1

Reviewer 1 Report

Audia and colleagues reviewed rare, but often difficult to manage hematological entity, Evans’ syndrome (ES) in this paper. Again, available prospective trials comparing treatment modalities are almost non-existent in this rare disorder, which makes proposing a treatment guidance is very difficult. The manuscript is a broad review, but needs some clarifications/improvements.

General/Major issues

The paper heavily focuses on the adult experience. Including differences or tendencies between pediatric and adult cases is, such as secondary or primary case incidences is useful whenever the data is available. There has to be wider referenced data included on genetic links of ES particularly in childhood studies, since it is very likely that these could be pertinent to primary/idiopathic ES cases in adulthood as well.

Throughout the manuscript, authors have appeared to have referred to ITP for thrombocytopenia and AIHA for anemia in ES cases, leading to some confusion. As exemplified in line 281, “… as monotherapy is low for ITP (20-30%) and for AHIA (33%) … “. It should be clarified, if they mean monotherapy fixed thrombocytopenia in patients with ES or they referred to monotherapy has been successful in isolated cases of ITP. If the former is what was meant, they should use “thrombocytopenia” instead of ITP or anemia/hemolytic anemia instead of AIHA here and throughout. In addition, separately, it has to be emphasized when  isolated ITP or isolated AIHA is meant in the text.

Either authors include details of pediatric-associated definitions, and data or include “adult” or “adulthood” in the title of the manuscript.

When the results/outcomes are provided for ES, the incidence ranges for certain components and different development courses from the published literature is useful; however, when referring to certain range values, the age range of the subjects and primary vs. secondary status should be included from the referred references whenever available.

Requires significant editing.

Diagnosis

Complement factor 3 and 4 levels and CH50 are also useful and need to be included in Table 1.

Line 125. … 35% of the patients had died. Clarify: They died from underlying disease vs. with active disease vs. while in remission for lymphoproliferative disorder.

Line 127. Use the complete names of disorders and in a standard manner throughout the manuscript: systemic lupus erythematosus.

Line 144. SARS-CoV-2 is the correct reference of this virus.

Line 282. The statement “neutropenia should not be considered a contraindication for corticosteroids in ES” is not necessary.

Line 376. The authors reviewed the EBMT data in the beginning of the paragraph; later focused on solely ES cases. At the end of the page, they went back to give results on the entire group without making it clear that they are back to reviewing the results on the whole group. The way the paragraph is organized is difficult to follow. Thus, it needs to be reorganized, maybe using paragraphs, etc.

Author Response

We are grateful to the reviewer for her/his comments that have helped us to improve our manuscript.

We are pleased to provide a point to point response:

  • Comment: The paper heavily focuses on the adult experience. Including differences or tendencies between paediatric and adult cases is, such as secondary or primary case incidences is useful whenever the data is available.

Response: to be in agreement with the different reviewers, a specific paragraph dedicated to paediatric has been written (paragraph 3) and the differences between adults and paediatric are mentioned (lines 561-563 and 576-577)

  • Comment: There has to be wider referenced data included on genetic links of ES particularly in childhood studies, since it is very likely that these could be pertinent to primary/idiopathic ES cases in adulthood as well.

Response: as required, publications related to genetic disorders associated with ES are now specifically detailed in paragraph 3.2. (line 589-684, references 85-100).

  • Comment: Throughout the manuscript, authors have appeared to have referred to ITP for thrombocytopenia and AIHA for anaemia in ES cases, leading to some confusion. As exemplified in line 281, “… as monotherapy is low for ITP (20-30%) and for AHIA (33%) … “. It should be clarified, if they mean monotherapy fixed thrombocytopenia in patients with ES or they referred to monotherapy has been successful in isolated cases of ITP. If the former is what was meant, they should use “thrombocytopenia” instead of ITP or anaemia/haemolytic anaemia instead of AIHA here and throughout. In addition, separately, it has to be emphasized when isolated ITP or isolated AIHA is meant in the text.

Response: we do agree that the formulation we used was confusing. To improve the comprehension of the manuscript, we now clearly mentioned isolated ITP or isolated AIHA when not dealing with ES, while  ITP during ES is referred to as ES-thrombocytopenia and AIHA during ES is referred to as ES-anaemia. This has been changed all along the manuscript.

  • Comment: Either authors include details of paediatric-associated definitions, and data or include “adult” or “adulthood” in the title of the manuscript.

Response: In agreement with recommendations of reviewers, ES in paediatrics is now included in this review (paragraph 3). Thus the title has been changed as appropriate to “Evans’ syndrome: from diagnosis to treatment”.

  • Comment: When the results/outcomes are provided for ES, the incidence ranges for certain components and different development courses from the published literature is useful. However, when referring to certain range values, the age range of the subjects and primary secondary status should be included from the referred references whenever available.

Response: As suggested, the range values of ages have been added (lines 68-69, 565 and 567).

  • Comment: Complement factor 3 and 4 levels and CH50 are also useful and need to be included in Table 1.

Response: we do agree with the reviewer that complement measurement is of interest during the diagnosis of isolated AIHA, particularly during cold agglutinin disease that is associated with complement consumption. However, considering the fact that ES- anaemia is a warm haemolytic anaemia, complement measurement is of low interest in this situation. However, as complement can decrease during lupus flare, its measurement is of interest in lupus-associated ES. This has been specified (lines 184-186).

  • Comment: Line 125. … 35% of the patients had died. Clarify: They died from underlying disease vs. with active disease vs. while in remission for lymphoproliferative disorder.

Response: it has now been specified that all the patient had an active haematological malignancy that was the cause of the death in 4/6. In only one case, ES participated to the death with a haemorrhagic stroke probably favoured by a partial response of thrombocytopenia (lines 148-153).

  • Comment: Line 127. Use the complete names of disorders and in a standard manner throughout the manuscript: systemic lupus erythematosus.

Response: Systemic lupus erythematosus (lines 72 and 166) is now used and abbreviated as SLE in all the manuscript.

  • Comment: Line 144. SARS-CoV-2 is the correct reference of this virus.

Response : Change has been made as recommended (line 191).

  • Comment: Line 282. The statement “neutropenia should not be considered a contraindication for corticosteroids in ES” is not necessary.

Response: we do agree with the reviewer that clinicians that are used to manage ES patients will not be reluctant to used steroids or immunosuppressants in case of ES neutropenia due to the underlying autoimmune mechanism. However, due to the fact that this review could be of interest for a broader range of  clinicians, we do thing that it is important to specify that steroids and immunosuppressants should not be considered a contraindication in ES neutropenia. This has been specifically specified lines 309-312.

  • Comment: Line 376. The authors reviewed the EBMT data in the beginning of the paragraph; later focused on solely ES cases. At the end of the page, they went back to give results on the entire group without making it clear that they are back to reviewing the results on the whole group. The way the paragraph is organized is difficult to follow. Thus, it needs to be reorganized, maybe using paragraphs, etc.

Response: as suggested, the paragraph has been reorganized to help distinguishing general results from those from ES when known (lines 397-428).

Reviewer 2 Report

In this review, Audie at al report an overview on diagnosis and treatment of a Evans' syndrome, a very rare and life-threatening event whose knowledge is crucial in clinical practice of hematologists.

 The manuscript is easily readable but its structure should be completely revised to better focus the most important diagnostic and therapeutic  points. Other important points need to be revised.

1) Overall Structure

-A very brief introduction should be added.

-Although at pag 5 line 146 the authors state that Pediatric ES will not be extensively detailed, they often underline specific issues that inevitably need to be highlighted  due to the differences with adult population. I would dedicate a paragraph  on Pediatric ES in order  to underline the specific underlying disorders in children (PIDs, ALPS other CIDs etc) and -above all – to avoid misunderstandings for treatment.

-Although the Title involves “complication and clinical management”, only few lines on complications are reported. I would enlarge this part (or change the title...). A specific paragraph on the management of life-threatening hemolysis or hemorrhages could be added at this regard.

-In the “Diagnostic procedures” paragraph, the section on underlying disorders should be limited to the specific diagnostic exams. The treatment approaches should be moved in the clinical management paragraph and could also be further clarified with 1 Table/Figure

2) The table should be more synthetic and most of explanation should be  moved to the text

3) Section 2.2.4.2:

References on diagnostic criteria of ALPS should be added and the role of MMF in  ALPS patients should be reported and referenced

 4) Section 3.1.1 Corticosteroids

Although the authors state that steroids are the  cornerstone therapy this section need to be more comprehensive. For instance, steroids should be also given at higher dosages (that should be specified ) in severe cases and the duration of treatment also better reported  for AIHA : “no longer” than six months is confusing since a slow six-months period is usually considered as an adequate decalage timing

5) Section 3.2.2 /3.2.3

Line 347: the response rate of all listed  immunosuppressant is given 56%. This makes no sense and should be better reported  highlighting  the role and data of  each drugs in both adult and pediatric section.

6) References are limited

Author Response

We are grateful to the reviewer for her/his comments that have helped us to improve our manuscript.

We are pleased to provide a point to point response:

Comment: The manuscript is easily readable, but its structure should be completely revised to better focus the most important diagnostic and therapeutic points.

Response: as requested, the structure of the article has been completely refined. There is now a clear distinction between adults and children ES (paragraphs 2 and 3, lines 48 and 556). The therapeutic points are also highlighted in Table 2 (line 296).

- Comment: A very brief introduction should be added.

Response: as recommended a brief introduction (paragraph 1, line 41)) defining ES is now provided. Data regarding epidemiology have been moved to specific sections for both adults (paragraph 2.1, line 59) and children (paragraph 3.1., line 557).

- Comment: Although at page 5 line 146 the authors state that Paediatric ES will not be extensively detailed, they often underline specific issues that inevitably need to be highlighted due to the differences with adult population. I would dedicate a paragraph on Pediatric ES in order to underline the specific underlying disorders in children (PIDs, ALPS other CIDs etc) and -above all – to avoid misunderstandings for treatment.

Response: as requested, a specific section is now specifically dedicated to ES in children (paragraph 3), reporting on ES epidemiology, genetic disorders associated with childhood ES, treatments and evolution (lines 556-685).

- Comment: Although the Title involves “complication and clinical management”, only few lines on complications are reported. I would enlarge this part (or change the title...). A specific paragraph on the management of life-threatening haemolysis or haemorrhages could be added at this regard.

Response: considering all the modifications of the previous manuscript, the title has been changed for “Evans’ syndrome: from diagnosis to treatment”. As recommended the management of life threating haemolysis and haemorrhages is now detailed (lines 494-553).

- Comment: In the “Diagnostic procedures” paragraph, the section on underlying disorders should be limited to the specific diagnostic exams. The treatment approaches should be moved in the clinical management paragraph and could also be further clarified with 1 Table/Figure

Response: As recommended, treatment approaches that were mentioned in the “diagnostic procedure” section were moved to the “treatment” section. A specific paragraph is dedicated to ES during pregnancy (lines 465-476).

- Comment: The table should be more synthetic and most of explanation should be moved to the text

Response: as recommended, table 1 (line 108) has been simplified and the explanations have been moved to the text in the appropriate sections (“determining the secondary nature of ES: lines 154-162; 181-186; 193-197).

- Comment: References on diagnostic criteria of ALPS should be added and the role of MMF in ALPS patients should be reported and referenced

Response: as recommended the diagnosis criteria for ALPS are now mentioned (lines 638-649) with the appropriate reference (ref 96: Oliveira et al. 2010). Similarly, the response to MMF is now mentioned (lines 655-657) with the appropriate quotation (ref 42: Miano et al. 2016).

- Comment: Section 3.1.1 Corticosteroids. Although the authors state that steroids are the cornerstone therapy this section need to be more comprehensive. For instance, steroids should be also given at higher dosages (that should be specified ) in severe cases and the duration of treatment also better reported  for AIHA : “no longer” than six months is confusing since a slow six-months period is usually considered as an adequate decalage timing

Responses: as recommended the duration of steroid treatment has been clarified. The possibility of using higher dosages (1.5 mg/kg) of prednisone (lines 301-306) or pulses of methylprednisolone is now specified (lines 306-307; 500-502; 537-538)

- Comment: Section 3.2.2 /3.2.3. Line 347: the response rate of all listed immunosuppressant is given 56%. This makes no sense and should be better reported highlighting the role and data of each drugs in both adult and paediatric section.

Response: as requested, the overall response to immunosuppressant is reported in the text (line 383), while the responses to each immunosuppressant are reported in table 2 (line 296).

- Comment: references are limited

Responses: the number of publications quoted has been increased (from 59 to 100) in link with the recommendations of reviewers. It needs to be acknowledged that the specific literature on ES is scarce due to the rarity of the disease. The major publications have been quoted and we did not quote all clinical cases that are of limited value in such a review.
